# Network-based restoration strategies maximize ecosystem recovery

Udit Bhatia [1,2,5 ✉], Sarth Dubey[3,5], Tarik C. Gouhier [4] & Auroop R. Ganguly[2]

Redressing global patterns of biodiversity loss requires quantitative frameworks that can predict ecosystem collapse and inform restoration strategies. By applying a network-based dynamical approach to synthetic and real-world mutualistic ecosystems, we show that biodiversity recovery following collapse is maximized when extirpated species are reintroduced based solely on their total number of connections in the original interaction network. More complex network-based strategies that prioritize the reintroduction of species that improve 'higher order' topological features such as compartmentalization do not provide meaningful performance improvements. These results suggest that it is possible to design nearly optimal restoration strategies that maximize biodiversity recovery for data-poor ecosystems in order to ensure the delivery of critical natural services that fuel economic development, food security, and human health around the globe.

[1] Discipline of Civil Engineering, Indian Institute of Technology, Gandhinagar, Gujarat 382355, India. [2] Sustainability and Data Sciences Lab, Department of Civil and Environmental Engineering, Northeastern University, Boston, MA 02115, USA. [3] Discipline of Computer Science and Engineering, Indian Institute of Technology, Gandhinagar, Gujarat 382355, India. [4] Department of Marine and Environmental Sciences, Marine Science Center, Northeastern University, Nahant, MA 01908, USA. [5] These authors contributed equally: Udit Bhatia, Sarth Dubey. ✉email: bhatia.u@iitgn.ac.in

Restoring degraded ecosystems is crucial in an era of rapid global change[1–3]. However, reversing declining trends in biodiversity and ecosystem functioning requires an understanding of how to preserve ecosystem integrity[4] or, if already severely degraded, restore species and their functions to their original state[5]. Mutualistic networks are particularly vulnerable to ecosystem degradation, as their stability depends on a set of strongly interdependent species and interactions[6,7]. The loss of even one species can cause a ripple effect and lead to secondary extinctions that compromise the entire system's stability[8,9]. These properties suggest that sequentially restoring the most critical species and their interactions in the network may promote recovery[10], strengthen resilience, and increase functioning[11]. The challenge lies in assessing the criticality of each species at each step of the restoration process in order to optimize recovery across different ecosystems.

In the past decade, there has been an increase in ecological restoration efforts around the globe[2,12]. Recent theoretical studies have demonstrated that complex ecosystems consisting of many interacting species exhibit a universal pattern of system collapse[13]. If ecosystems tend to collapse in the same manner, it is reasonable to expect a similar universal pattern of recovery, a phenomenon that could be used to design optimal restoration strategies. However, it remains to be seen whether a single universal restoration strategy can be generalized across disparate ecosystems to maximize multiple key criteria such as persistence, total species abundance, and faster stabilization after species reintroduction. Here, we seek to determine whether such a generalizable restoration strategy can be developed based on the network topology of ecosystems and the underlying dynamics of interacting species following perturbations of different magnitudes[14,15].

Although returning a severely degraded ecosystem to its original state can be difficult, it remains a key goal for many conservation and restoration projects[12,16]. Traditional attempts to design optimal restoration strategies have focused on single-species[17], suitable habitat identification[18], prioritization of sites that maximize spatial rescue effects in communities interconnected by dispersal[19], or have used low-dimensional models with a few interacting components[20]. More recently, researchers have highlighted the need to shift from single species to entire interaction networks in order to design effective restoration strategies[21,22], with the expectation that the most effective recovery strategy may involve reversing the sequence of species loss that led to the maximum amount of habitat loss or number of secondary extinctions[23,24]. Other approaches have used phylogenetic relationships to address the restoration of mutualistic ecosystems such as plant-pollinator networks in data-scarce regions[25]. For food web networks, researchers have used measures of topological centrality as indicators of species importance or 'keystoneness' in the context of extinction. Although these centrality indicators are strongly correlated, they yield divergent prioritization schemes or rankings for species reintroductions that ultimately influence the effectiveness of restoration strategies[15]. Hence, there is a lack of consensus about how to quantify a species' role in restoration gains when ecosystems undergo varying magnitudes of degradation, and to what degree 'keystone' species contribute to biodiversity gains within complex networks. Therefore, a generic framework for identifying an effective multispecies restoration strategy that accounts for the topological and dynamical characteristics of high-dimensional ecological networks remains elusive. Below, we address this critical knowledge gap by determining whether effective restoration approaches that account for the full species interaction network can be generalized to maximize the recovery of geographically and topologically diverse ecosystems while ensuring stable dynamics of multispecies assemblages.

## Results and discussion

The dynamical behavior of ecosystems, including mutualistic networks, is typically simulated by high-dimensional equations that capture how species interact with each other and their environment. However, the characterization of such systems in multi-dimensional parameter space often results in intractable and unpredictable behavior. Recently, researchers have proposed formalisms to reduce the complexity of these n-dimensional systems where n represents the number of species in 1-D[13] and 2-D[26] systems of equations. While these frameworks offer analytical insights into the collapsing behavior of these systems when subjected to external perturbations, they have yet to be adopted to design restoration strategies.

**Integrating network topology and ecological dynamics for mutualistic networks.** Here, we combine network topology measures with the underlying ecological dynamics simulated through a 1-D model that is simple to analyze but does not account for mutualistic interactions comprehensively; a 2-D model that captures the bipartite and mutualistic nature of interactions with one variable representing aggregate plants and another for aggregate pollinators; and an n-dimensional coupled dynamical model that explicitly accounts for all species.

We conducted a systematic analysis of 30 real-world plant-pollinator mutualistic systems around the globe to identify a universal and near-optimal restoration strategy (Fig. 1a; Supplementary Table 1). To cover the spectrum of ecological properties that these real-world networks do not capture, we also simulated 27 synthetic networks with varying attributes (Supplementary Fig. 1, Supplementary Table 2). We compared the different network-based and random (null) restoration strategies via three key criteria measured after the reintroduction of each species: abundance $X$[13], settling time $ST$[6], and persistence $P$[27]. Specifically, we examined whether restoration strategies based on attributes measured by network topology systematically resulted in meaningful gains in abundance while ensuring persistence and faster stabilization (i.e., lower settling time) relative to null restoration strategies based on random species reintroduction sequences.

**Perturbation scenarios and network-based restoration strategies.** Figure 1b (Supplementary Data 1) shows a representative mutualistic network generated synthetically and the ripple effect of secondary extinctions triggered by a single primary extinction (Fig. 1c, Supplementary Data 1). Each node represents individual species in the complex network representation, whereas the links represent the mutualistic relationships between a pair of plants and pollinators. In our analysis, we reduce the abundance of a perturbed species under consideration to zero and obtain the new equilibrium states using the three models described above (i.e.,1-D, 2-D, and n-dimensional models (Supplementary Fig. 2)). We then track the proportion of surviving species (persistence), abundances of surviving species, and time taken for abundance to reach the equilibrium state (a measure of settling time, quantified using n-dimensional model). Since the loss of species does not always happen randomly in real-world ecosystems, we constructed three perturbation scenarios: generalists preferred (probability of primary extinction was directly proportional to the number of mutualistic interactions or degree of the node), specialists preferred (probability of primary extinction was inversely proportional to the number of mutualistic interactions or degree of a node), and random (non-strategic removal of species).

Post perturbation, we used network-based restoration strategies to identify the most critical species at each step of the recovery process. Specifically, at each time step, we restored the most 'critical' species based on its topological importance in the

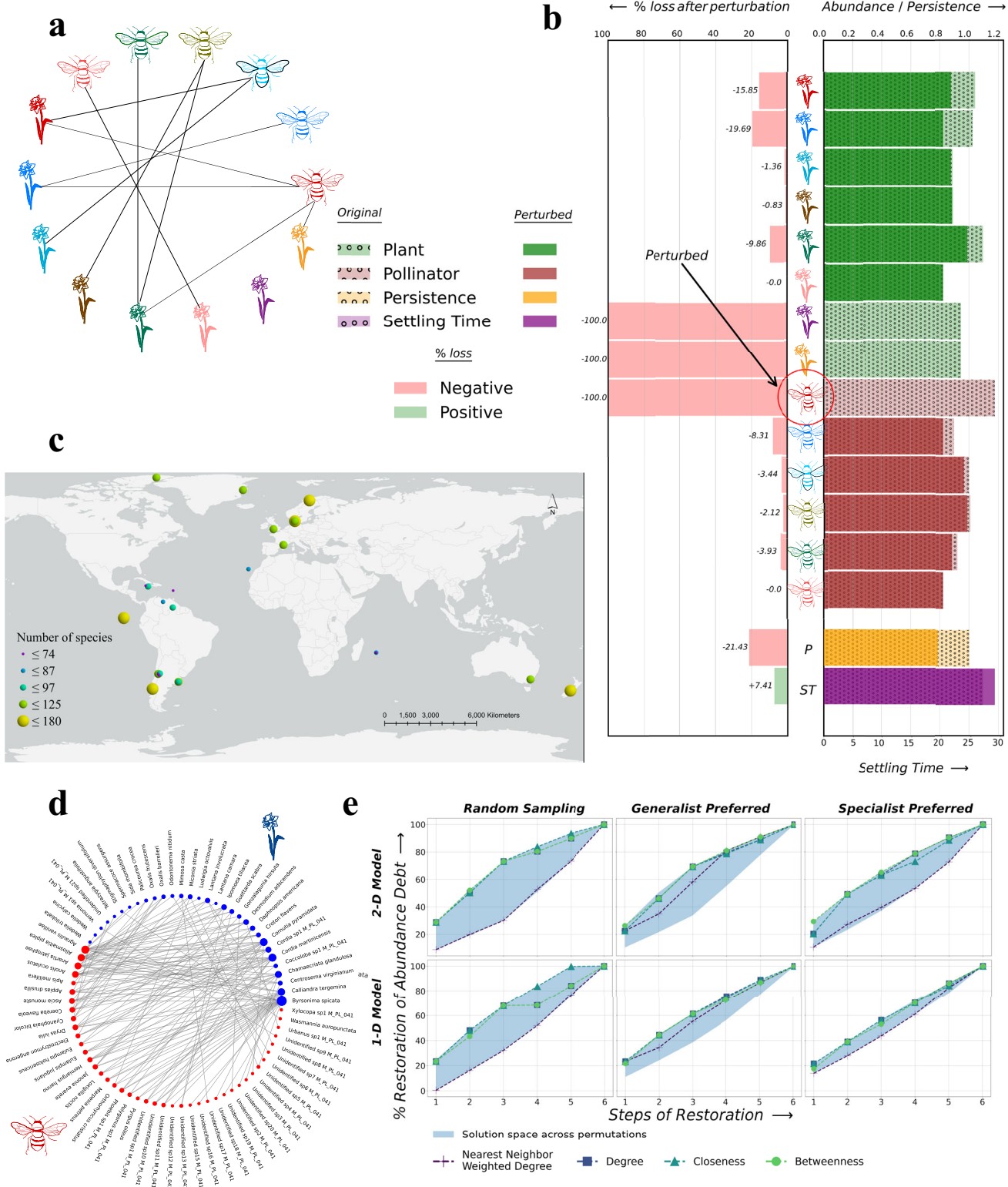

**Fig. 1 Illustration of network-informed restoration strategies for mutualistic systems. a** Geographical and ecological information about the 30 real-world mutualistic interaction networks. **b** A representative mutualistic network generated synthetically **c** is used to show the secondary extinctions that a primary extinction trigger. **d** An example ecosystem (`M_PL_041` in Supplementary Table 1) is perturbed by removing 20% of the plant species (6 species). **e** Outcomes measured in terms of percentage mean abundance and persisting species recovered for restoration strategies based on each topological property are plotted over the solution space generated by all possible reintroduction sequences.

network as captured by each of the following three centrality measures: (i) degree (a 'first order' metric based solely on each species' number of mutualistic interactions), (ii) closeness centrality (a 'higher order' metric that measures the proximity of each species to others in the network based on the pattern of interactions), and (iii) betweenness centrality (a 'higher order' metric that measures the degree to which species act as bridges in the network by linking otherwise unconnected subsets of species[28]). For each of these three network-based restoration strategies, we first created a degraded ecosystem by removing species using three perturbation scenarios. We then used 1-D, 2-D, and n-dimensional dynamical equations to simulate the system response after each species reintroduction.

We focused on network-based restoration strategies because they can be applied to any mutualistic network with known interactions[15,28], but with limited or no information on interaction strengths, parameters controlling competitive and mutualistic dynamics, and phylogenetic relationships[25]. Hence, such approaches have the potential to yield system-agnostic and context-independent restoration strategies for data-poor systems. Furthermore, to understand the relationship between network architecture and recovery from perturbations, we measured the following attributes: network size (total number of species), asymmetry (ratio of pollinators to plants), connectance (proportion of possible links realized), and nestedness (defined by the disassortative assemblage of interactions)[29]. These attributes have been linked to the resilience of mutualistic and non-mutualistic networks[6]. Hence, assessing their relationships with species abundance ($X$), settling time ($ST$), and persistence ($P$) can help us determine the suitability of recently popularized dimensionality reduction approaches for guiding restoration[13,26].

**Simulating recovery dynamics for an example ecosystem.** To illustrate this approach, we used one of the 30 real-world networks (Fig. 1d) as an example ecosystem. This ecosystem is located in Syndicate, Dominica, and it comprises 74 species (hereafter referred to as nodes), with 31 plants and 43 pollinator species. Before simulating the system's response to reintroduction, we remove 20% of plant species (6 species) using all three scenarios. However, for real-world ecosystems, we considered 30%, 60%, and 90% of node removal under all three perturbation scenarios (9 combinations in total). Even with a small fraction of the nodes removed (20% in this case) for the example ecosystem, there are 6! or 720 possible restoration pathways when these species are introduced sequentially. While changes in abundance, persistence, and time to stabilization are tractable exhaustively for such networks (Fig. 1e for abundance; Supplementary Fig. 3a for settling time, and Supplementary Fig. 3b for persistence), prioritization becomes a challenge for species-rich ecosystems undergoing massive degradation. For both 1-D and 2-D models, the region of all possible 720 reintroduction pathways is drawn for all three perturbation scenarios with 20% of nodes removed (Fig. 1e, Supplementary Data 1). Overall, reintroduction sequences based on species degree, closeness, and betweenness offer near-optimal mean abundance recovery for each perturbation scenario. The nearest-neighbor average degree achieves similar performance for generalist-preferred perturbation but not for random and specialist-preferred perturbation scenarios.

For this example network, while strategies that prioritize species based on their degree, closeness, or betweenness centralities result in relatively faster gains in abundance (Fig. 2a and b, Supplementary Data 2), random restoration of species can result in faster stability (Fig. 2c and d), at the cost of species persistence (Fig. 2e and f) for both 1-D and 2-D models. Fig. 2 shows the comparative analysis of various restoration strategies

for the generalist-preferred perturbation scenario with perturbation corresponding to 60% of the species being removed (see Supplementary Fig. 4–11 for the remaining eight perturbation combinations). We observed a significant difference in the distribution of mean abundance based on topology-driven restoration strategies compared to random (or non-strategic) interventions ($p < 0.05$, 2-sample Kolmogorov–Smirnov test; Supplementary Table 3). Given the strong correlation between degree and betweenness (Spearman Corr = 0.95; $p < 0.05$), no significant difference is observed in the distributions of abundance between strategies where restoration prioritizes degree vs. betweenness for both 1-D and 2-D models.

**Identifying "winning" strategies for species reintroduction across 30 real-world ecosystems.** To determine whether these results hold for diverse ecosystems located around the globe that exhibit large variations in their network structures (Supplementary Fig. 1a), we simulated the nine combinations of perturbation scenarios for all 30 real-world networks and quantified the efficacy of each prioritization strategy by simultaneously measuring the three criteria mentioned above (i.e., abundance, settling time, persistence). Despite broad similarities in the performance of distinct restoration strategies that echo those found in the example ecosystem, we found clear and important differences in their effectiveness for each criterion across both 1-D and 2-D models.

Figure 3a summarizes the "winning" reintroduction strategies for both the 1-D and 2-D models for each of the three criteria of interest (i.e., abundance, settling time, and persistence) across all 30 real-world networks (see Supplementary Fig. 12 for a similar analysis of the 27 synthetic networks). For both 1-D and 2-D models, centrality-guided restoration emerged as the most effective strategy to recover abundance for ecosystems with disparate sizes and complexities undergoing varying magnitudes of perturbations under all three removal scenarios. A similar pattern was observed for persistence, where betweenness-guided restoration emerged as the "winner" in 90.37% of the simulations, and other topological centralities accounted for the remaining 9.63% (Fig. 3b, Supplementary Data 3). However, for the 1-D model, betweenness-guided restoration was the most efficient "winning" strategy across 37.04% of the simulations, followed by closeness-guided restoration at 34.81%. In contrast to the "winning" strategies for maximizing mean abundance, the random species reintroduction strategy yielded the lowest time to stabilization in 58.52% of the simulations. This is because random restoration results in fewer species that persist in the network resulting in faster stabilization on average. However, the comparison of the reintroduction strategies on an example ecosystem (Fig. 1d) reveals minor differences in relative performances between network-based restoration strategies (see Supplementary Figs. 13 and 14 for similar analysis on all 30 real-world networks).

The analysis for 2-D models showed that although betweenness-guided restoration outperformed the other strategies, with degree-guided restoration being a close second, the performance difference between degree-guided and betweenness-guided restoration strategies was within 1% for all criteria (Supplementary Fig. 13). Similarly, the performance difference between closeness- and betweenness-guided restoration was within 1% for 7.78%, 18.52% and 5.56% of the simulations for the three criteria (i.e., abundance, settling time, and persistence), respectively (see Supplementary Fig. 15). Furthermore, the performance difference between random and betweenness-guided restoration strategies was 1% or less in 1.11% of the simulations for settling time (see Supplementary Fig. 16). The

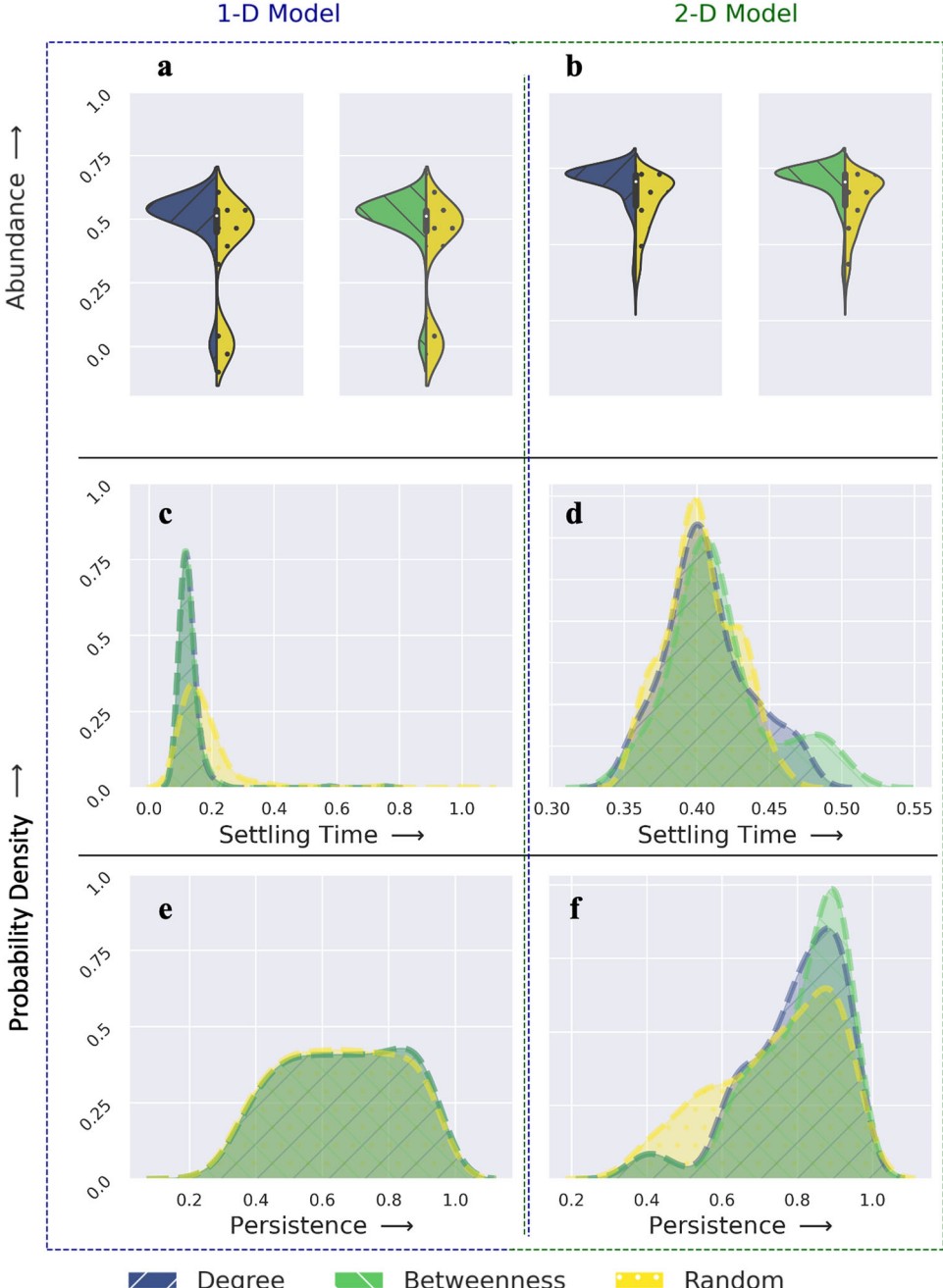

**Fig. 2 Performance of different restoration strategies for the example mutualistic network.** Distributions of mean abundance X (**a**, **b**), settling time ST (**c**, **d**), and persistence P (**e**, **f**) for each restoration strategy under generalist-preferred perturbation scenarios where 60% of all species were removed.

small performance differences between the network-based strategies is likely due to their strong correlations across real-world networks (see Supplementary Fig. 17 for aggregated results across all 30 networks and Supplementary Fig. 18 for pairwise correlations for each of the 30 real-world networks).

**Relationships between network attributes and restoration criteria.** As noted earlier, the 1-D model is simple to analyze but does not account for mutualistic interactions comprehensively, whereas the 2-D model fully captures the bipartite and mutualistic nature of the interactions with one collective variable for plants and another for pollinators. Although using these dimensionality reduction models to understand the robustness, resilience, and restoration pathways is pragmatic, it is important to

determine whether their predictions concur with those of the full n-dimensional system. Specifically, do the 1-D and 2-D projection approaches preserve the relationships between network measures and species mean abundance ($X$), settling time ($ST$), and persistence ($P$) observed in the full n-dimensional networks? In our analysis, we kept track of the criteria and the structural properties of the ecosystem network (i.e., network size, asymmetry, connectance, and nestedness) during the restoration process and then computed the Spearman rank correlation between the criteria and the structural properties for both the 1-D and 2-D models.

Figure 4a shows the relationship between the mean abundance and the established network structural properties for the 1-D and 2-D models (see Supplementary Fig. 19a for settling time, and Supplementary Fig. 19b for persistence; Supplementary

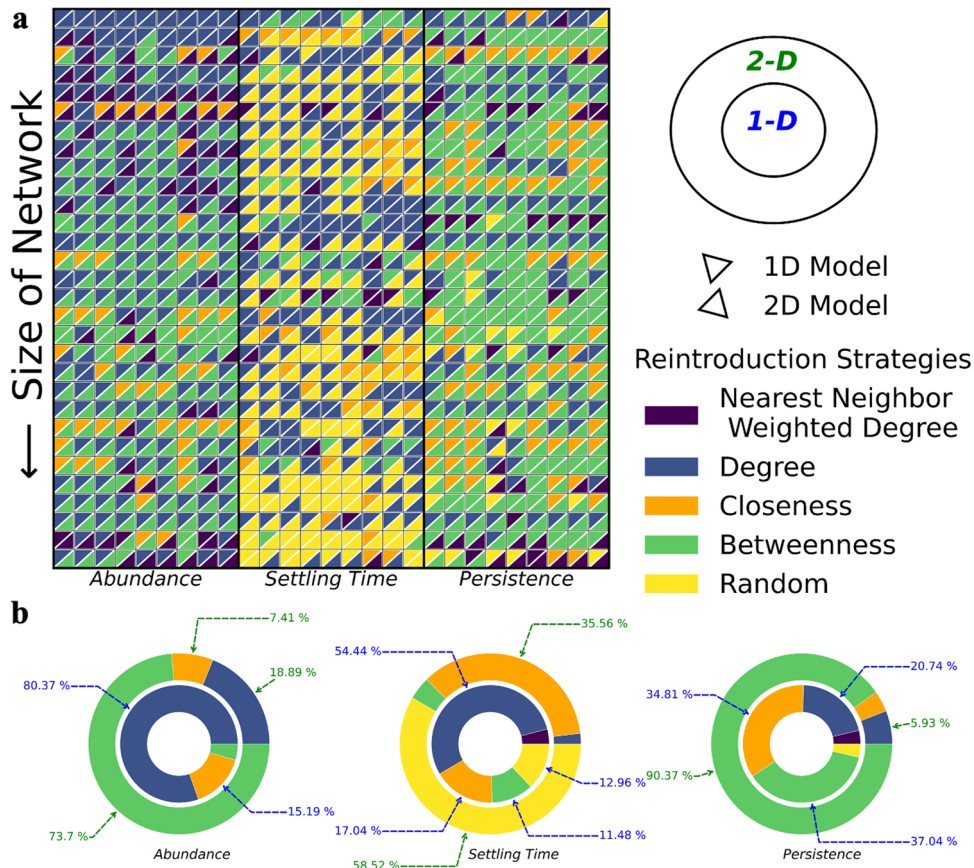

**Fig. 3 Identifying the most effective restoration strategies for each criterion. a** For the 30 real-world networks studied with the three criteria of interest (blocks with nine columns each) and all nine perturbation scenarios: random selection of species (columns 1–3), generalist-preferred (columns 4–6) or specialist-preferred (columns 7–9), with perturbation corresponding to the removal of 30%, 60%, and 90% species (ordered left to right for each perturbation scenario). Restoration strategies are ranked based on a `best-vs-rest' policy for 1-D and 2-D models (upper and lower triangle, respectively). **b** The distribution of the best restoration strategies is drawn in a nested pie chart (1-D and 2-D models on inner and outer regions, respectively) for each criterion.

Figs. 20–22 for the real-world and synthetic networks studied). In the 2-D model, we observed a negative relationship between mean abundance and network size and asymmetry but a positive relationship between mean abundance and either connectance or nestedness. This is consistent with the ecological literature, where higher connectance and nestedness have been linked to more stable mutualistic ecosystems. However, although the 1-D model results are consistent with the reported statistical relationships between connectance and nestedness, the relationship reverses for network size and asymmetry. The positive relationship between abundance and network size observed in the 1-D model does not agree with the literature and thus shows the potential limits of projection schemes that fail to capture the full complexity of natural ecosystems when restoring perturbed mutualistic systems. Figure 4b shows the relationship between the criteria used to measure the effectiveness of the restoration strategies. The highest mean abundance was systematically associated with lower settling time and lower persistence in both the 1-D and 2-D models. Therefore, faster stabilization and greater mean abundance are realized at the cost of fewer species persisting within the ecosystem, which is a suboptimal and undesirable outcome. This tradeoff supports our choice of using multiple criteria for assessing the effectiveness of species restoration strategies in perturbed mutualistic systems.

**Revealing why 'first order' network-based strategies are near-optimal.** Overall, our findings indicate that more complex 'higher

order' network approaches where species prioritization is based on betweenness or closeness centrality provide only minimal improvements over 'first order' strategies based solely on species connectivity or generality. That is, the benefits of targeting more intricate 'higher order' effects by reintroducing species that optimize network compartmentalization or nestedness–two attributes associated with the stability and persistence of ecological networks–offer marginal restoration gains[6]. Instead, reintroducing species based on their 'first order' effects as measured by generality is sufficient to ensure an effective restoration strategy because generality is usually correlated with 'higher order' network effects (see Supplementary Fig. 17).

While the relationship between a species' generalism and its contribution to nestedness–a measure of organization within mutualistic networks–seems weak when viewed collectively, as depicted in Supplementary Figs. 17 and 18, a detailed analysis uncovers a robust positive correlation between nestedness and generalism during the initial phases of ecological restoration. Notably, the greatest enhancement in nestedness occurs during the first restoration steps when species are removed based on generalist-preferred criteria and reintroduced through network-centric approaches (See Supplementary Fig. 23). By prioritizing the reintroduction of generalist species (or using other centrality measures strongly correlated with generality), we achieve near-optimal gains in species abundance and persistence, particularly at the outset of restoration when nestedness improvements are most pronounced. This aligns with the notion that nestedness fosters

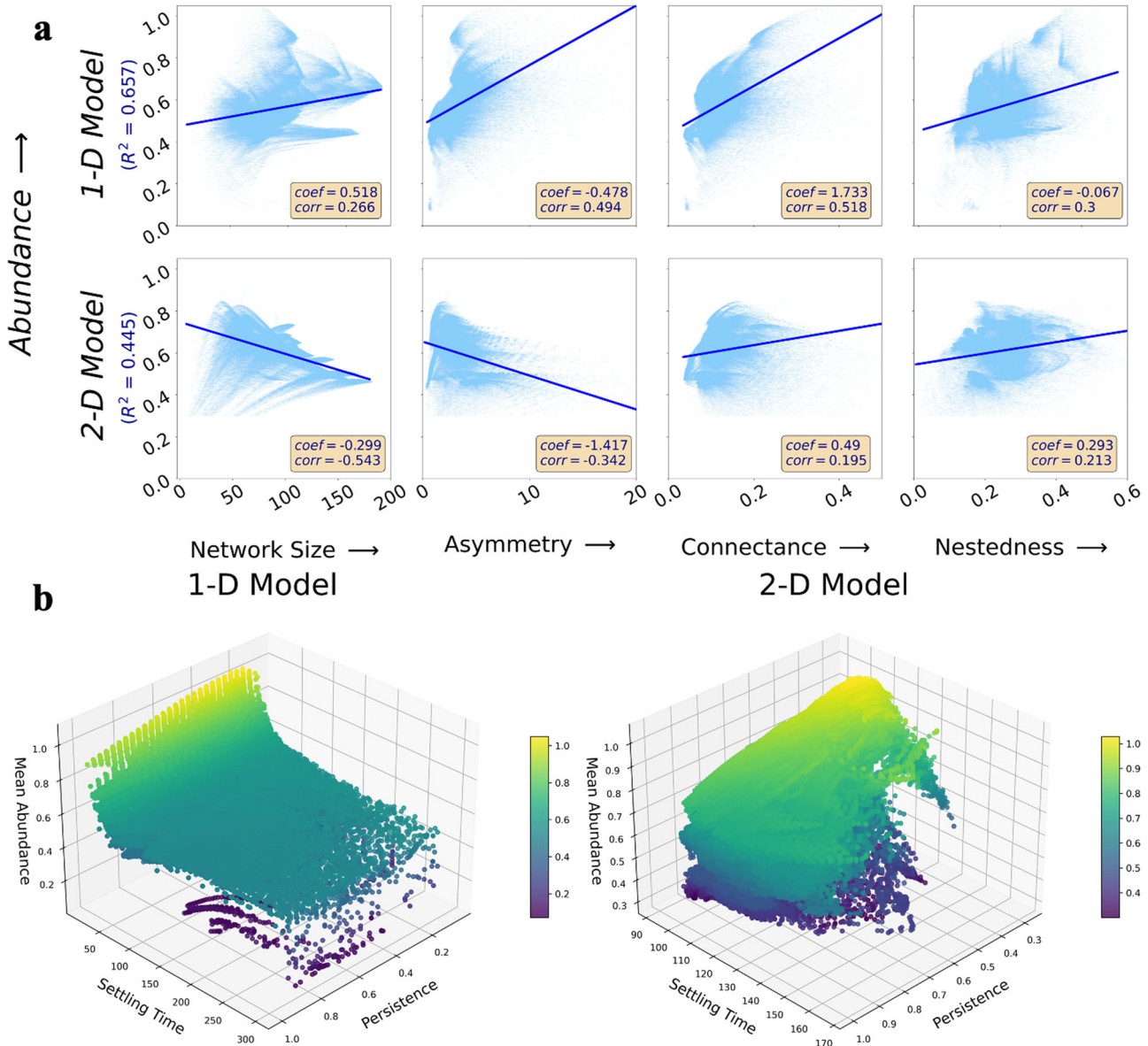

**Fig. 4 Restoration performance is correlated with network topology. a** The mean abundance of the surviving species obtained after each reintroduction for every perturbed scenario considered. The panels include the Spearman rank correlation ('corr') as well as both the coefficient of multiple linear regression ('coef') and the coefficient of multiple determination $R^2$ using normalized values of the network structural properties. **b** Surface plots show the relationships between the three criteria (mean abundance, settling time, and persistence) for both the 1-D and 2-D models. The plot reveals the trade-offs between maximizing abundance and persistence while reducing settling time when designing restoration strategies for mutualistic networks using dimensionality reduction approaches.

stability and resilience within networks[6]. It is crucial to recognize that in mutualistic networks, only a small percentage of species significantly contribute to overall nestedness. Consequently, over time, changes in nestedness contributions become marginal, which may partially account for the observed weak yet meaningful correlations at the aggregate level. Another factor potentially influencing this weak correlation is that species with varying degrees of centrality could have similar marginal contributions to total nestedness, further obscuring the relationship at the aggregate level.

## Conclusions

Trait-based approaches offer valuable insight into the design of effective restoration strategies. However, obtaining the necessary data for such analyses is a considerable challenge, particularly due to the vast geographical, phylogenetic, and environmental

disparities that exist between ecosystems[30]. In this study, we have analyzed 30 real-world ecosystems around the globe that are characterized by differences sizes, environmental conditions, and ecological attributes. We demonstrated that using network properties such as species connectivity or generalism to guide restoration efforts can assist in identifying near-optimal pathways for recovering abundance while ensuring persistence. Additionally, the strong and significant correlations observed between 'first order' characteristics such as species connectivity or generality and other 'higher order' network properties suggest that the choice of a specific centrality measure for designing restoration strategies may not be critical, as the differences between the most effective and the next-best network solutions are generally within a 1% margin for all criteria across all ecosystems and scenarios.

Conservation efforts often focus on habitat restoration because much of the observed biodiversity loss can be attributed to habitat

loss and fragmentation[31]. Given the widely observed species-area relationships, such habitat-based restoration efforts are likely to lead to positive outcomes (all things being equal). Here, we wanted to focus on an alternate premise: how can we promote biodiversity recovery in ecosystems suffering from local species extirpation following a strong (pulse) environmental perturbation[32] such as a prolonged heat wave or a persistent cold snap when habitat restoration is neither necessary (e.g., because the environmental perturbation has not influenced habitat availability) nor possible (e.g., due to financial or spatial management constraints)? Our results thus represent an important "worst case" scenario establishing the kinds of results one can expect under different network-based restoration strategies when more traditional approaches, such as habitat amelioration are not available. Our results also highlight the complex relationships among three criteria that are often used to assess the effectiveness of restoration strategies: abundance, settling time, and persistence. Although low settling time may suggest rapid stabilization, this outcome is often associated with weak long-term persistence and lower overall species abundance. For example, achieving rapid stabilization may come at the cost of fewer species persisting in the system, thus eroding overall resilience. Conversely, an exclusive focus on abundance could negatively impact settling time and long-term persistence. Our study highlights the importance of carefully assessing and prioritizing these three interdependent criteria when designing restoration strategies for entire ecosystems.

Many have proposed that the sixth mass extinction is well underway and is mainly attributable to direct and indirect anthropogenic effects[33,34]. Global change has severely disrupted critical mutualistic ecological networks[35], with climate-mediated shifts in temperature expected to exacerbate extinction risks for insects, including pollinators[32]. These changes have far-reaching impacts on plant diversity, ecosystem stability, and even crop production. In this context, our study offers insights into optimal restoration strategies that may be particularly useful under varying environmental conditions. Although we focused primarily on designing restoration strategies to remediate the effects of species loss, our framework can easily be adapted to address a wider range of relevant perturbations in conservation biology. Future work could extend our approach to include the impact of temporal environmental variability and thus allow the development of adaptive restoration strategies that mitigate the impacts of environmental change in 'real-time'. Addressing these challenges is of utmost importance for the preservation of the planet's interconnected ecosystems.

## Methods

**Mutualistic plant-pollinator networks**. We analyzed 30 mutualistic plant-pollinator networks (listed in Supplementary Table 1) from the Web of Life database (www.web-of-life.es), representing them as unweighted, undirected bipartite Species Interaction Networks (SINs). In these networks, nodes represent species, and edges depict mutualistic interactions between a pair of nodes. Further, we measure the following attributes for these SINs: Size (S): Total number of species in the mutualistic network. That is, $S = n + m$; where $n$ and $m$ represent the number of plant and pollinator species, respectively; Asymmetry (A): The ratio of pollinator to plant species, defined as $A = \frac{m}{n}$; Connectance (C): The proportion of possible links that are realized. That is, $C = \frac{L}{(m \cdot n)}$; where $L$ is the total number of interactions; Nestedness (N): The degree to which the species interactions are structured in a hierarchical way. It measures the extent to which species with fewer interactions are subsets of species with more interactions. The NODF metric ranges from 0 to 100, where 0 indicates a

completely non-nested matrix, and 100 indicates a perfectly nested matrix[26].

**Simulating dynamics**. The dynamical models simulate species abundances within plant-pollinator networks by incorporating processes such as intrinsic growth, intraspecific and interspecific competition, and mutualistic interactions between pollinators and plants.

*n-D model*. Mathematically, the n-D model is represented as follows:

$$\frac{\partial P_i}{\partial t} = P_i\left(\alpha_i^{(P)} - \sum_{j=1}^{S_P} \beta_{ij}^{(P)} P_j + \frac{\sum_{k=1}^{S_A} \gamma_{ik}^{(P)} A_k}{1 + h\sum_{k=1}^{S_A} \gamma_{ik}^{(P)} A_k}\right) + \mu^{(P)} \quad (1)$$

$$\frac{\partial A_i}{\partial t} = A_i\left(\alpha_i^{(A)} - \sum_{j=1}^{S_A} \beta_{ij}^{(A)} A_j + \frac{\sum_{k=1}^{S_P} \gamma_{ik}^{(A)} P_k}{1 + h\sum_{k=1}^{S_P} \gamma_{ik}^{(A)} P_k}\right) + \mu^{(A)} \quad (2)$$

Here, $P_i$ and $A_i$ represent the abundances of the $i$th plant and pollinator species, respectively, with $S_P$ and $S_A$ denoting the total number of plant and pollinator species within the network. The parameter $\alpha$ represents the intrinsic growth rate in the absence of competition and mutualistic effects, while $\beta_{ii}$ and $\beta_{ij}$ (for $i \neq j$) characterize intraspecific and interspecific competition respectively; $\mu$ parametrizes the immigration of species; and $h$ is the half-saturation constant. The parameter $\gamma$ quantifies the strength of mutualistic interactions, with $\gamma = 0$ indicating the absence of such interactions in the network. Generally, $\gamma$ is dependent on the degree (i.e., number of mutualistic partners) of species $i$ ($D_i$), as follows:

$$\gamma_{ij} = \epsilon_{ij}\frac{\gamma_0}{(D_i^p)} \quad (3)$$

where $\gamma_0$ is a constant, $\epsilon_{ij} = 1$ if there is an interaction between species $i$ and $j$ or 0 otherwise, $p$ characterizes the tradeoff between the interaction strength and the number of mutualistic links.

*2-D Model*. The n-dimensional model (eq. (1) and eq. (2)) can be dimensionally reduced to a 2-D model in terms of effective abundance of plant ($P_{eff}$) and pollinator ($A_{eff}$) species as:

$$\frac{\partial P_{eff}}{\partial t} = \alpha P_{eff} - \beta P_{eff}^2 + \frac{<\gamma_P> A_{eff}}{1 + h<\gamma_P> A_{eff}}P_{eff} + \mu \quad (4)$$

$$\frac{\partial A_{eff}}{\partial t} = \alpha A_{eff} - \beta A_{eff}^2 + \frac{<\gamma_A> P_{eff}}{1 + h<\gamma_A> P_{eff}}A_{eff} + \mu \quad (5)$$

The values of all parameters are listed in Supplementary Table 4. Physical interpretations of each parameter, along with detailed derivations of equations (4) and (5), are available in Jiang et al.[26].

*1-D model*. Given that mutualistic interactions occur between a plant and a pollinator species, and not within the plant or pollinator species themselves, the formed network is bipartite in nature. Consequently, we can project this bipartite network onto plants or pollinators to simulate species abundances, using the projected adjacency matrix ($A_{ij}$)[36].

Mathematically, we represent the n-dimensional model employing the projected adjacency as follows:

$$\frac{\partial x_i}{\partial t} = B_i + x_i\left(1 - \frac{x_i}{K_i}\right)\left(\frac{x_i}{C_i} - 1\right) + \sum_{j=1}^{N} A_{ij}\frac{x_i x_j}{D_i + E_i x_i + H_j x_j} \quad (6)$$

In this model, $x_i$ denotes the abundance of the $i$th plant or pollinator species, contingent on the chosen projection of the adjacency. Incorporated processes in this model include constant

rate incoming migration $B_i$, logistic growth governed by a carrying capacity $K_i$, Allee's effect, which results in reduced growth at lower abundances ($x_i < C_i$), and the positive effects of mutualistic interactions between species $i$ and $j$. These interactions are influenced by the projected adjacency $A_{ij}$ and are parameterized by $D_i$, $E_i$, and $H_j$.

Using the methodology delineated in Gao et al.[13], we simplify eq. (6) to a one-dimensional model (eq. (7)) to quantify the effective abundance, $x_{eff}$, of plants and pollinators:

$$\frac{\partial x_{eff}}{\partial t} = B + x_{eff}\left(1 - \frac{x_{eff}}{K}\right)\left(\frac{x_{eff}}{C} - 1\right) + \beta_{eff}\frac{x_{eff}^2}{D + (E + H)x_{eff}}$$

(7)

Here, $\beta_{eff}$ characterizes the nearest-neighbor weighted degree of the projected network. The values of all parameters are listed in Supplementary Table 4. Physical interpretations of each parameter, along with detailed derivations of equation (7), are available in Gao et al.[13].

**Simulating perturbations**. In this study, the nodes correspond to individual species, while the edges delineate the mutualistic interactions between pairs of plant and pollinator species. Within this framework, a perturbation is defined as the elimination of a species, represented by removing the corresponding node and its associated edges from the network. We impose a model of obligate mutualism: a species is deemed extinct, and consequently it, along with its connecting edges is excised from the network if it loses all mutualistic partners. Following the removal of a specific node and its links, we update the adjacency matrix, which can be either the projected or the original matrix, depending on the dimensionality of the model. We use three distinct perturbation scenarios to select the species to be perturbed. The first, random sampling, involves the random removal of species. The second, generalist-preferred sampling, entails selecting species with a probability directly proportional to their degree, which reflects the number of mutualistic partners they possess. On the contrary, the third scenario, specialist-preferred sampling, selects species with a probability inversely proportional to their degree. Each of these perturbation scenarios is applied to the removal of 30%, 60%, and 90% nodes within a plant-pollinator network, thus generating nine unique combinations in total. To mitigate statistical biases, we analyze 10-ensembles for each combination.

**Key criteria**. To evaluate the relative efficacy of different recovery strategies in our dynamical simulations, we monitor the following key criteria: Abundance ($X$)[13]: The total number of individuals of a species in an ecosystem.; Settling Time ($ST$)[6]: The time taken by the dynamical models to achieve a steady state for every species. A steady state is confirmed when the population abundance of each species remains within a predefined tolerance limit of $10^{-6}$ for at least five-time steps; and Persistence ($P$)[27]: Calculated as the proportion of species that survive (i.e., those not going extinct) to the total number of species in the network, once a steady state is achieved.

**Simulating recovery**. Post perturbation, we simulate the ecosystem's recovery by strategically reintroducing species deemed most 'critical' to the perturbed network. This reintroduction process not only brings back the selected species, but also restores their respective mutualistic links within the network. During each step of this restoration process, the three key criteria are recalculated using dynamical models. The criterion for determining the 'criticality' of species hinges on their topological significance within the network, gauged by four distinct measures[13,28]: Nearest-neighbor weighted degree ($\beta_{eff}$), Degree Centrality, Closeness Centrality, and Betweenness Centrality. For centrality-based measures, the species with the highest centrality score is prioritized for reintroduction, being identified as the most 'critical'. In contrast to these centrality measures, $\beta_{eff}$ yields a single value characterizing the entire perturbed network. Consequently, the sequence of species reintroduction is determined by selecting species in a manner that maximizes the incremental increase in $\beta_{eff}$ at each restoration step. To evaluate the efficacy of these reintroduction strategies, we used a null model in which species are restored in a random sequence until the network achieves complete recovery.

**Identifying 'winning' strategies**. We subject each network to perturbations using the 9 combinations of scenarios, with 10 ensembles for each combination. To simulate species abundances using dynamical models, we set an initial abundance of $10^{-6}$ for each species. These simulations are conducted for a maximum of 300 time steps, equivalent to a span of 100 years, with each time step representing a period of 4 months. This temporal framework enables detailed monitoring of the three key criteria in each restoration step.

To determine the most effective restoration strategy for each criterion, we calculate the average value of each criterion over the course of the restoration steps and across all ensembles. This approach not only accounts for variations in the length of restoration sequences, but also ensures that the contributions of different species at various stages are duly considered. This averaging is crucial for comparing networks of different sizes, as a uniform perturbation percentage (e.g., 30%) results in the removal of varying numbers of nodes depending on the network size. Additionally, the average of ensembles addresses potential statistical biases arising from sampling. The 'winning' strategy is determined based on its performance against these criteria: higher values of abundance ($X$) and persistence ($P$), and lower values of settling time ($ST$) are deemed superior. In instances where there is a tie in a key criterion, we resort to the other criteria (abundance and settling time, in that order) to designate the 'winner'.

**Statistics and reproducibility**. We investigated the linear relationship between each of the three key restoration criteria and the network attributes, assessing how these relationships are affected by 1-D and 2-D models. For each criterion, we normalize both the criterion and the network attributes on a scale of 0–1, based on their respective maximum and minimum values. Using simple linear regression (with a sample size of $N \sim 10^5$), we regress each criterion against the attribute of the network. Subsequently, we determined the Spearman Rank Correlation for each best-fit line derived from both 1D and 2D models. Furthermore, we perform a multivariate linear regression, in which each criterion is regressed against all attributes of the network collectively, and calculate $R^2$, the ratio of the variance explained to the total variance. To ensure the reproducibility of the results, the dynamical equations and the values of the parameters and time steps are provided in the Methods and Supplementary Table 4, and the associated codes are provided via Zenodo repository[37]. Source data underlying figures 1–3 are provided in Supplementary Data 1–3, respectively.

**Reporting summary**. Further information on research design is available in the Nature Portfolio Reporting Summary linked to this article.

**Data availability**
Data for 30 mutualistic plant-pollinator networks were obtained from the Web of Life database (www.web-of-life.es). Data used to generate the figures are available in Zenodo[37].

## Code availability

The computer code for analyzing the data and creating the graphs was written in Python. The code along with the datasets can be downloaded from Zenodo[37].

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

## Acknowledgements

This research was funded by the Indian Institute of Technology Gandhinagar startup grant and Indian MoE STARS #367, the US NSF Cyber SEES (1442728), the US NSF BIG DATA (#1447587), and the US NSF Expeditions in Computing (1029711) grants, in addition to an ongoing US DOD Strategic Environmental Research and Development Program (SERDP) funding (# RC20-1183). Other sources of funding were NSF SES-1735505 for A.R.G. and NSF OCE-2048894 for T.C.G., respectively. Additional support for A.R.G. comes from the AI for Climate and Sustainability focus area within the Northeastern University (NU) Institute for Experiential AI (IEAI), headquartered in Boston, MA, USA, and the Roux Institute in Portland, ME, USA. The work was initiated when U.B. was a Ph.D. student at the NU Sustainability and Data Sciences Laboratory (SDS Lab) in Boston, MA, USA. Data were obtained from http://www.web-of-life.es developed by Jordi Bascompte's lab.

## Author contributions

Conceptualization: U.B., T.G., A.R.G. Methodology: U.B., S.D., T.G., A.R.G. Analysis: U.B., S.D. Visualization: U.B., S.D. Writing—original draft: UB, SD. Writing—review and editing: U.B., S.D., T.G., A.R.G. U.B. and S.D. contributed equally.

## Competing interests

The authors declare no competing interests.
