## [Peer review file · Communications Biology]

Reviewers' comments:

Reviewer #1 (Remarks to the Author):

This study applied a network-based dynamical approach to synthetic and real-world mutualistic ecosystems, showing that biodiversity recovery following collapse is maximized when extirpated species are reintroduced based solely on their total number of connections in the original interaction network. This study displays the optimal restoration strategies for biodiversity recovery. In general, this work is novel, and the method sounds. In particular, this topic is timely, as ecosystem restoration is becoming a widely recognized solution to the biodiversity crisis. However, I have two major points that require the authors to clarify:

(i) They have ignored a significant conservation action - habitat management for biodiversity recovery, meaning that the authors only focused on species re-introduction for mutualistic ecosystem recovery while ignoring spatial habitat restoration. This makes this study less realistic. In other words, conservation ecologists prefer habitat restoration over species re-introduction for biodiversity maintenance, as environmental change might influence the outcome. As such, this work is too ideal to apply to real ecosystems. Thus, I would like to ask the authors to justify this point, or whether they can integrate habitat restoration into the mechanistic model.

(ii) Recently there is one important literature relating to this topic: Gawecka, K. A., & Bascompte, J. (2023). Habitat restoration and the recovery of metacommunities. *Journal of Applied Ecology*, 00, 1–15. <https://doi.org/10.1111/1365-2664.14445>

I would like to ask the authors to clarify their novelty compared to this work, since in my mind, this work has more important practical implications.

Overall, I'd ask the authors to address the two major points above before I can recommend for publication in *Communications Biology*.

Reviewer #2 (Remarks to the Author):

In this study, the authors aim to investigate optimal restoration strategies for degraded ecosystems, focusing specifically on plant-pollinator mutualistic networks. The authors argue that effective restoration approaches should prioritize the network level rather than the species level due to the lack of consensus regarding the ranking of species importance under context dependency, such as keystone-ness and species contribution. As a result, the authors propose a network-informed

approach and simulate 1-D (no interaction), 2-D (one aggregated plant-pollinator interaction), and n-dimensional models (all species interactions). The basic idea behind this network-informed approach is to restore the most topologically critical species in terms of degree, closeness, or betweenness at each step of the restoration process. Subsequently, the authors apply this approach to 27 synthetic and 30 real-world plant-pollinator ecosystems, introducing three types of perturbations (i.e., species removal): generalists preferred, specialists preferred, and random. The authors then evaluate the winning restoration strategy by assessing three criteria that involve trade-offs: maximizing (1) the proportion and (2) the abundance of surviving species and (3) minimizing the time required for abundance to reach equilibrium. Furthermore, the authors conclude that knowledge of the original interactions, i.e., network degree or ecological species connectivity and generality, is sufficient to maximize biodiversity recovery, indicating a limited need for higher-order topological features like closeness and betweenness.

Upon reviewing the manuscript, I am convinced that the restoration strategy for mutualistic ecosystems holds significant importance, as the extinction of a single species can rapidly lead to secondary extinctions and even the collapse of the entire ecosystem. The authors have performed an innovative and systematic topological analysis aimed at comparing and identifying the optimal restoration strategy. Additionally, the insight that the near-optimal strategy requires minimal information is intriguing, particularly for restoring data-poor ecosystems. However, I believe that the manuscript can be further enhanced by addressing the following major and minor points.

Major points:

1. I agree with the authors' emphasis on comparing the predictions derived from 1-D and 2-D models with those from n-dimensional systems. Also, in lines 74-78, the authors mention the simulations of n-dimensional coupled dynamical systems. However, throughout the main text and supporting material, the only implicit mention of the n-dimensional model that I found is in line 203, where the authors claim that the patterns generated by the 2-D model aligns with the ecological literature (without references). I believe this comparison should be more explicit. I suggest reporting the simulation results of the n-dimensional models alongside the 1-D and 2-D models, either within the text or using figures.

2. Figure 2 requires a detailed description as it currently appears confusing. For instance, in Panels A and B, why does the y-axis "abundance" scale from 0 to 1? How does it differ from the y-axis "density" in Panels C, D, E, and F, which also ranges from 0 to 1? Do they have the same meaning in this figure? Furthermore, regarding the caption of Panels A and B, does "mean abundance" correspond to the average abundance of all species in the ecosystem (also seen in Figure 4)? If so, for the example ecosystem in Figure 2, why does "mean abundances" have a distribution rather than a single value? I found in the supporting material that "abundance" is a measure of relative frequency or proportion of each species within an ecosystem, which is different from the classical definition. How does this relate to "mean abundance"? In Panels C and D, why does the "settling time" (i.e., the time taken to achieve steady state) range from 0 to 1? Does this correspond to the average "settling time" for every species in the ecosystem? How can we conclude that "random restoration of species

can result in faster stability" (lines 143-144)? If this conclusion is based on the peaks of the distributions, it seems that only in the 2-D model (Panel D), random restoration of species can be slightly faster (i.e., smaller settling time). Again, related to point 1, a systematic comparison and reporting of the results from the n-dimensional model would be beneficial.

Minor comments:

1. In Figure 3, what does the "system" reintroduction strategy represent? In the supporting material, I found that it may correspond to the "nearest-neighbor average degree". If so, how is "nearest" quantified? Is it β_{eff} (as shown in Table S4)? But in Eq. S7, β_{eff} also means the strength of mutualistic interactions between species. Are they the same?

2. In line 170, there appears to be a typo: the restoration at 34.81% refers to closeness-guided restoration rather than degree-guided restoration.

3. In the Supporting Material -> Method -> 3. Connectance (C), the formula is missing a parenthesis, which should be $C = L/(m \cdot n)$.

4. The title of this manuscript mentions maximizing ecosystem recovery, and I believe it would be valuable to discuss the overall impact of the three criteria (abundance, settling time, and persistence) on ecosystem recovery, especially in light of the trade-offs.

5. In the last section, the authors mention global environmental change and the mass extinction. How does the study address the influence of environmental factors on the ecosystem restoration process? Are there specific considerations for how environmental changes, beyond species removal, can impact the efficacy of restoration efforts?

6. Regarding restoration, Prof. Andrew Gonzalez's work is worth checking, such as <https://www.pnas.org/doi/abs/10.1073/pnas.2211288120?doi=10.1073/pnas.2211288120>.

Response to Reviewer Comments on COMMSBIO-23-1744-T, Titled "Network-based restoration strategies maximize ecosystem recovery."

Response to Reviewers:

Reviewer #1 (Remarks to the Author):

This study applied a network-based dynamical approach to synthetic and real-world mutualistic ecosystems, showing that biodiversity recovery following collapse is maximized when extirpated species are reintroduced based solely on their total number of connections in the original interaction network. This study displays the optimal restoration strategies for biodiversity recovery. In general, this work is novel, and the method sounds. In particular, this topic is timely, as ecosystem restoration is becoming a widely recognized solution to the biodiversity crisis. However, I have two major points that require the authors to clarify:

(i) They have ignored a significant conservation action - habitat management for biodiversity recovery, meaning that the authors only focused on species re-introduction for mutualistic ecosystem recovery while ignoring spatial habitat restoration. This makes this study less realistic. In other words, conservation ecologists prefer habitat restoration over species re-introduction for biodiversity maintenance, as environmental change might influence the outcome. As such, this work is too ideal to apply to real ecosystems. Thus, I would like to ask the authors to justify this point, or whether they can integrate habitat restoration into the mechanistic model.

Thank you for your constructive feedback. This is an excellent point. Conservation efforts often focus on habitat restoration because much of the observed biodiversity loss can be attributed to habitat loss and fragmentation. Given the widely observed species-area relationships, such habitat-based restoration efforts are likely to lead to positive outcomes (all things being equal). Here, we wanted to focus on a more interesting problem: how can we promote biodiversity recovery in ecosystems suffering from local species extirpation following a strong (pulse) environmental perturbation such as a prolonged heat wave or a persistent cold snap when habitat restoration is neither necessary (e.g., because the environmental perturbation has not influenced habitat availability) nor possible (e.g., due to financial or spatial management constraints)? Our results thus represent an important "worst case" scenario establishing the kinds of results one can expect under different network-based restoration strategies when more traditional approaches such as habitat amelioration are not available.

Additionally, our results provide an important baseline that can be used to benchmark the performance of restoration strategies based on habitat amelioration. We have added text to the discussion to articulate these key points and help frame our results in the context of more traditional habitat-based restoration strategies. Overall, we believe that our results complement existing habitat-based approaches and thus thank the reviewer for prompting us to articulate these points more clearly in the discussion [*Lines 255-264; 268-275*].

(ii) Recently there is one important literature relating to this topic: Gawecka, K. A., & Bascompte, J. (2023). Habitat restoration and the recovery of metacommunities. *Journal of Applied Ecology*, 00, 1– 15. <https://doi.org/10.1111/1365-2664.14445>

I would like to ask the authors to clarify their novelty compared to this work, since in my mind, this work has more important practical implications.

Thank you for calling attention to this important paper, which was first published after our paper was submitted for review. The Gawecka and Bascompte paper focused on restoration strategies for metacommunities (i.e., communities occupying different patches or sites that are interconnected by dispersal). As such, their results identified the spatial strategies that maximized the recovery of interconnected antagonistic vs mutualistic communities. This is very different from our study, which focuses on identifying the temporal strategies that maximize the recovery of isolated communities. Specifically, the Gawecka and Bascompte study focused on the optimal sequence of sites to restore, relying on dispersal and spatial rescue effects to allow biodiversity to recover in disturbed sites. We, on the other hand, simulated the worst case scenario whereby no spatial rescue effect was possible (due to lack of dispersal) and identified the optimal sequence of species reintroductions in order to promote biodiversity recovery. Hence, our temporal approach and results complement those presented in Gawecka and Bascompte. We have added text to the manuscript to emphasize the differences between our results and those presented in Gawecka and Bascompte [**Lines 47-49**].

Reviewer #2 (Remarks to the Author):

In this study, the authors aim to investigate optimal restoration strategies for degraded ecosystems, focusing specifically on plant-pollinator mutualistic networks. The authors argue that effective restoration approaches should prioritize the network level rather than the species level due to the lack of consensus regarding the ranking of species importance under context dependency, such as keystone-ness and species contribution. As a result, the authors propose a network-informed approach and simulate 1-D (no interaction), 2-D (one aggregated plant-pollinator interaction), and n-dimensional models (all species interactions). The basic idea behind this network-informed approach is to restore the most topologically critical species in terms of degree, closeness, or betweenness at each step of the restoration process. Subsequently, the authors apply this approach to 27 synthetic and 30 real-world plant-pollinator ecosystems, introducing three types of perturbations (i.e., species removal): generalists preferred, specialists preferred, and random. The authors then evaluate the winning restoration strategy by assessing three criteria that involve trade-offs: maximizing (1) the proportion and (2) the abundance of surviving species and (3) minimizing the time required for abundance to reach equilibrium. Furthermore, the authors conclude that knowledge of the original interactions, i.e., network degree or ecological species connectivity and generality, is sufficient to maximize biodiversity recovery, indicating a limited need for higher-order topological features like closeness and betweenness.

Upon reviewing the manuscript, I am convinced that the restoration strategy for mutualistic ecosystems holds significant importance, as the extinction of a single species can rapidly lead to secondary extinctions and even the collapse of the entire ecosystem. The authors have performed an innovative and systematic topological analysis aimed at comparing and identifying the optimal restoration strategy. Additionally, the insight that the near-optimal strategy requires minimal information is intriguing, particularly for restoring data-poor ecosystems. However, I believe that the manuscript can be further enhanced by addressing the following major and minor points.

We appreciate your thoughtful review and your recognition of the significance of our research on restoration strategies for mutualistic ecosystems. Your positive remarks on our systematic topological analysis and the insight regarding near-optimal strategies in data-poor ecosystems are especially encouraging. We have addressed the major and minor points you have highlighted to enhance the quality and impact of our manuscript.

Major points:

1. I agree with the authors' emphasis on comparing the predictions derived from 1-D and 2-D models with those from n-dimensional systems. Also, in lines 74-78, the authors mention the simulations of n-dimensional coupled dynamical systems. However, throughout the main text and supporting material, the only implicit mention of the n-dimensional model that I found is in line 203, where the authors claim that the patterns generated by the 2-D model aligns with the ecological literature (without references). I believe this comparison should be more explicit. I

suggest reporting the simulation results of the n-dimensional models alongside the 1-D and 2-D models, either within the text or using figures.

We appreciate your suggestion to make a clearer comparison between the 1-D, 2-D, and n-dimensional models. To address this, we have added the missing references to support our findings. Additionally, simulation results for the n-dimensional model related to our sample network are now included in the Supplementary Information, as Figure S2. We did not include n-dimensional model outputs for all ecosystems in the main text because of the complexity of showing the outputs for large ecosystems. Specifically, the number of metrics can become overwhelming when calculated as n multiplied by the number of species. To manage this, we included only summary statistics in the final plots. However, to fully address your concern, the complete output of the n-dimensional model and the 1-D and 2-D models for our sample network is now available in the Supplementary Information [**new fig. S2**].

2. Figure 2 requires a detailed description as it currently appears confusing. For instance, in Panels A and B, why does the y-axis "abundance" scale from 0 to 1? How does it differ from the y-axis "density" in Panels C, D, E, and F, which also ranges from 0 to 1? Do they have the same meaning in this figure? Furthermore, regarding the caption of Panels A and B, does "mean abundance" correspond to the average abundance of all species in the ecosystem (also seen in Figure 4)? If so, for the example ecosystem in Figure 2, why does "mean abundances" have a distribution rather than a single value? I found in the supporting material that "abundance" is a measure of relative frequency or proportion of each species within an ecosystem, which is different from the classical definition. How does this relate to "mean abundance"? In Panels C and D, why does the "settling time" (i.e., the time taken to achieve steady state) range from 0 to 1? Does this correspond to the average "settling time" for every species in the ecosystem? How can we conclude that "random restoration of species can result in faster stability" (lines 143-144)? If this conclusion is based on the peaks of the distributions, it seems that only in the 2-D model (Panel D), random restoration of species can be slightly faster (i.e., smaller settling time). Again, related to point 1, a systematic comparison and reporting of the results from the n-dimensional model would be beneficial.

Thank you for the excellent comment. We address each component of your comment below:

Figure 2 requires a detailed description as it currently appears confusing. For instance, in Panels A and B, why does the y-axis "abundance" scale from 0 to 1?

We acknowledge the need for more clarity in the description of Figure 2, particularly concerning the y-axis scaling in Panels A and B. In our reduced dimension models, species abundances may vary across scales depending on the ecosystem studied. To facilitate a direct comparison of these outputs across multiple ecosystems, we normalized the abundance values between 0 and 1. Importantly, this normalization process keeps the integrity and interpretability of our results, which are focused on comparing the relative efficacy of various network restoration strategies for each of the reduced dimension models. Therefore, the y-axis in Panels A and B

scales from 0 to 1. To avoid confusion, we have renamed the y-axis as "Probability Density"
[Revised Figure 2; Figures S4-S11]

How does it differ from the y-axis "density" in Panels C, D, E, and F, which also ranges from 0 to 1? Do they have the same meaning in this figure? Furthermore, regarding the caption of Panels A and B, does "mean abundance" correspond to the average abundance of all species in the ecosystem (also seen in Figure 4)? If so, for the example ecosystem in Figure 2, why does "mean abundances" have a distribution rather than a single value?

We appreciate your inquiry concerning the nuances of Figure 2, which portrays the outcomes when a generalist-preferred 60% species loss perturbs the network. In Panels A and B, the "mean abundance" captures the average abundance of species across multiple realizations for this perturbed network. The "mean abundance" distribution arises from measuring these averages at each step during restoration across the 10 different realizations of our perturbed sample network. Conversely, in Panels C, D, E, and F, the y-axis labeled "density" is fundamentally different. It delineates the probability density of the observed settling times and persistence levels across the different realizations and restoration strategies. While the criteria have been normalized to fall within a 0 to 1 scale for easy comparison, the meaning ascribed to these y-axis labels differs across the panels.

I found in the supporting material that "abundance" is a measure of relative frequency or proportion of each species within an ecosystem, which is different from the classical definition. How does this relate to "mean abundance"?

Thank you for highlighting the definition of "abundance" as presented in the supporting material. We have revised the definition to align with the classical ecological understanding, specifying it as 'The total number of individuals of a species in an ecosystem.' In this context, the term "species abundance" is derived from the outputs of our reduced-dimension models, which consider and represent the biomass of each species in the ecosystem. This should clarify the relationship between "abundance" and "mean abundance" as used in our study
[Supplementary Information: Lines 105-111].

In Panels C and D, why does the "settling time" (i.e., the time taken to achieve steady state) range from 0 to 1? Does this correspond to the average "settling time" for every species in the ecosystem?

In our study, "settling time" is the time until every species in the ecosystem reaches a steady state. We have scaled this time to fit between 0 and 1 to make it easier to compare across different networks. We have explained it clearly in the revised SI **[Supplementary Information: Lines 112-114; 123-126].**

How can we conclude that "random restoration of species can result in faster stability" (lines 143-144)? If this conclusion is based on the peaks of the distributions, it seems that only in the

2-D model (Panel D), random restoration of species can be slightly faster (i.e., smaller settling time).

It is crucial to note that the assertion that 'random restoration can result in faster stability' should be interpreted as a possibility rather than a definitive outcome. This is substantiated by data from Figure 2 for our sample network, where the settling time in a randomly restored ecosystem is comparable to that in topology-based strategies. This is not just based on the peak of the distribution of settling times but also quantitatively affirmed in Figure S14, which shows the percentage performance based on min-max normalization, keeping in mind that lower settling time is better. Moreover, in several real-world networks, such as M_PL_028, 029, 034, 018, 019, 058, and 072_03, the random restoration strategy outperforms other methodologies across various perturbation scenarios. However, while the settling time may be lower, this could occur at the expense of reduced species persistence.

Minor comments:

1. In Figure 3, what does the "system" reintroduction strategy represent? In the supporting material, I found that it may correspond to the "nearest-neighbor average degree". If so, how is "nearest" quantified? Is it β_{eff} (as shown in Table S4)? But in Eq. S7, β_{eff} also means the strength of mutualistic interactions between species. Are they the same?

We have amended the terminology to clarify its meaning. We now refer to it as a 'nearest neighbor weighted degree' (β_{eff}) based strategy as defined in Gao et al. (2016). To elaborate on the calculation method: At each restoration step, $\beta_{\text{effective}}$ is calculated based on the existing species in the network. Subsequently, we assess the impact on $\beta_{\text{effective}}$ when reintroducing each species one at a time. The species whose reintroduction results in the maximum change in $\beta_{\text{effective}}$ is then selected for reintroduction into the network. This iterative process continues until the network is fully restored. We have implemented these clarifications and consistently revised the terminology in both the main text and the supporting information (SI) to minimize potential confusion [**Supplementary Information Lines 64-68; 87-88; Table S4**].

2 and 3. In line 170, there appears to be a typo: the restoration at 34.81% refers to closeness-guided restoration rather than degree-guided restoration. In the Supporting Material -> Method -> 3. Connectance (C), the formula is missing a parenthesis, which should be $C = L/(m \cdot n)$.

Thank you for pointing this out. We have fixed the typos [**Line 173**].

4. The title of this manuscript mentions maximizing ecosystem recovery, and I believe it would be valuable to discuss the overall impact of the three criteria (abundance, settling time, and persistence) on ecosystem recovery, especially in light of the trade-offs.

This is an excellent point. We have added the following paragraph in the revised manuscript to address this:

“Our results also highlight the complex relationships among three criteria that are often used to assess the effectiveness of restoration strategies: abundance, settling time, and persistence. Although low settling time may suggest rapid stabilization, this outcome is often associated with weak long-term persistence and lower overall species abundance. For example, achieving rapid stabilization may come at the cost of fewer species persisting in the system, thus eroding overall resilience. Conversely, an exclusive focus on abundance could negatively impact settling time and long-term persistence. Our study highlights the importance of carefully assessing and prioritizing these three interdependent and criteria when designing restoration strategies for entire ecosystems” [Lines 268-275].

5. In the last section, the authors mention global environmental change and the mass extinction. How does the study address the influence of environmental factors on the ecosystem restoration process? Are there specific considerations for how environmental changes, beyond species removal, can impact the efficacy of restoration efforts?

We have included a discussion in the last section to address this critical comment:

Many have proposed that the sixth mass extinction is well underway and mainly attributable to direct and indirect anthropogenic effects. Global change has severely disrupted critical mutualistic ecological networks, with climate-mediated shifts in temperature expected to exacerbate extinction risks for insects, including pollinators. These changes have far-reaching impacts on plant diversity, ecosystem stability, and even crop production. In this context, our study offers insights into optimal restoration strategies that may be particularly useful under varying environmental conditions. While we focused primarily on designing restoration strategies to remediate the effects of species loss, our framework can easily be adapted to address a wider range of perturbations relevant in conservation biology. Future work could extend our approach to include the impact of temporal environmental variability and thus allow the development of adaptive restoration strategies that mitigate the impacts of environmental change in ‘real-time’. Addressing these challenges is of utmost importance for the preservation of the planet’s interconnected ecosystems [Lines 277-287].

6. Regarding restoration, Prof. Andrew Gonzalez’s work is worth checking, such as <https://www.pnas.org/doi/abs/10.1073/pnas.2211288120?doi=10.1073/pnas.2211288120>.

Thank you for directing us to this recent contribution. This work, which establishes the persistence of sub-networks as a probabilistic indicator for the resilience of the entire

ecosystem, is indeed pertinent. We note that this work highlights the need to act under the limited availability of information and uses it as a probabilistic indicator. However, our framework adds a layer of complexity by recommending that multiple criteria, including abundance, settling time, and persistence, be considered concurrently to optimize overall ecosystem recovery. Our research also contributes to understanding restoration processes by leveraging reduced-dimension models and comparing the implications of first-order versus higher-order network properties. We will make sure to reference this insightful work in the revised manuscript ***[New Ref. 30; Line 246]***.

REVIEWERS' COMMENTS:

Reviewer #2 (Remarks to the Author):

The manuscript has been revised effectively in response to the feedback. I have no additional comments to offer.

Response to Reviewer Comments on COMMSBIO-23-1744A, Titled "Network-based restoration strategies maximize ecosystem recovery."

REVIEWERS' COMMENTS:

Reviewer #2 (Remarks to the Author):

The manuscript has been revised effectively in response to the feedback. I have no additional comments to offer.

We appreciate your time and effort in reviewing our manuscript and are pleased to receive your positive feedback on the revisions made.